# Antimicrobial Activity, Antioxidant Potential, Cytotoxicity and Phytochemical Profiling of Four Plants Locally Used against Skin Diseases

**DOI:** 10.3390/plants8090350

**Published:** 2019-09-15

**Authors:** John A. Asong, Stephen O. Amoo, Lyndy J. McGaw, Sanah M. Nkadimeng, Adeyemi O. Aremu, Wilfred Otang-Mbeng

**Affiliations:** 1Indigenous Knowledge Systems (IKS) Centre, Faculty of Natural and Agricultural Sciences, North West University, Private Bag X2046, Mmabatho 2735, South Africa; johnmilan058@gmail.com; 2Agricultural Research Council, Vegetables and Ornamental Plants, Pretoria, Private Bag X293, Pretoria 0001, South Africa; AmooS@arc.agric.za; 3Phytomedicine Programme, Department of Paraclinical Sciences, University of Pretoria, Private Bag X04, Onderstepoort, Pretoria 0110, South Africa; lyndy.mcgaw@up.ac.za (L.J.M.);; 4Food Security and Safety Niche Area, Faculty of Natural and Agricultural Sciences, North West University, Private Bag X2046, Mmabatho 2735, South Africa; 5School of Biology and Environmental Sciences, Faculty of Agriculture & Natural sciences, University of Mpumalanga, Private Bag X11283, Mbombela 1200, South Africa; Wilfred.mbeng@ump.ac.za

**Keywords:** antioxidant, antibacterial, antifungal, flavonoids, minimum inhibitory concentration, phenols, gas chromatography-mass spectrometry (GC-MS)

## Abstract

Although orthodox medications are available for skin diseases, expensive dermatological services have necessitated the use of medicinal plants as a cheaper alternative. This study evaluated the pharmacological and phytochemical profiles of four medicinal plants (*Drimia sanguinea*, *Elephantorrhiza elephantina*, *Helichrysum paronychioides*, and *Senecio longiflorus*) used for treating skin diseases. Petroleum ether and 50% methanol extracts of the plants were screened for antimicrobial activity against six microbes: *Bacillus cereus*, *Shigella flexneri*, *Candida glabrata*, *Candida krusei*, *Trichophyton rubrum* and *Trichophyton tonsurans* using the micro-dilution technique. Antioxidant activity was conducted using 2,2-diphenyl-1-picryhydrazyl (DPPH) free radical scavenging and β-carotene linoleic acid models. Cytotoxicity was determined against African green monkey Vero kidney cells based on the 3-(4,5-dimethylthiazol-2-yl)-2,5-diphenyltetrazolium bromide (MTT) colorimetric assay. Spectrophotometric and Gas Chromatography-Mass Spectrometry (GC-MS) methods were used to evaluate the phytochemical constituents. All the extracts demonstrated varying degrees of antimicrobial potencies. *Shigella flexneri*, *Candida glabrata, Trichophyton rubrum* and *Trichophyton tonsurans* were most susceptible at 0.10 mg/mL. In the DPPH test, EC_50_ values ranged from approximately 6–93 µg/mL and 65%–85% antioxidant activity in the β-carotene linoleic acid antioxidant activity model. The phenolic and flavonoid contents ranged from 3.5–64 mg GAE/g and 1.25–28 mg CE/g DW, respectively. The LC_50_ values of the cytotoxicity assay ranged from 0.015–5622 µg/mL. GC-MS analysis revealed a rich pool (94–198) of bioactive compounds including dotriacontane, benzothiazole, heptacosane, bumetrizole, phthalic acid, stigmasterol, hexanoic acid and eicosanoic acid, which were common to the four plants. The current findings provide some degree of scientific evidence supporting the use of these four plants in folk medicine. However, the plants with high cytotoxicity need to be used with caution.

## 1. Introduction

Skin diseases contribute significantly to the public health burden and continue to affect people of all ages. The increased rate of skin infections associated with immune compromised persons due to the human immunodeficiency virus (HIV) has put skin problems in the spotlight. Additionally, they constitute about 34% of occupational health problems encountered. Secondary symptoms of some infections such as syphilis manifest on the skin, impact negatively on the self-esteem and emotions of affected persons [1,2]. In rural areas, conditions such as poor hygiene, improper sanitation, lack of clean water, humid conditions and density of households are known to exacerbate the spread of skin diseases [3]. Despite the presence of conventional drugs for skin diseases, challenges such as unwanted side effects and development of resistance by microorganisms as well as the high cost of medications have prompted the shift of attention towards natural sources for remedies against skin diseases [4,5].

Skin diseases are attributed to a variety of causes and pathogens, of which the most commonly encountered ones are bacterial and fungal species. Bacteria often involved in skin infections include *Clostridium perfringens*, *Proteus mirabilis*, *Pseudomonas aeruginosa*, *Staphylococcus aureus* and *Streptococcus pyogenes* as well as bacteriodes which have been associated with diseases such as cellulitis, *Cancrumoris erysipelas*, erythrasma, impetigo, folliculitis, furuncles, abscesses, leprosy, secondary syphilis, yaws and ulcers [4,5]. Examples of fungi commonly and clinically associated with skin conditions include *Trichophyton rubrum*, *Candida albicans*, *Trichophyton tonsurans*, *Candida krusei* and *Candida glabrata*. These organisms are known to cause diseases such as athlete’s foot, mycids, ringworm (Tinea), pityriasis, candidiasis, mycetoma or madura foot [6,7].

Indigenous people use medicinal plants for treating skin diseases based on knowledge accumulated over-time and the biological efficacies of these plants have been attributed to the presence of diverse secondary metabolites such as flavonoids and phenols [8,9]. Additionally, medicinal plants serve as a source of bioactive compounds of pharmacological importance for humans [9,10,11,12]. Thus, the importance of scientifically testing the efficacy of plants (documented in folk medicine) in biological assays (for example, antimicrobial, antioxidant and anti-inflammatory activities) related to skin diseases and their safety evaluation has been well-emphasized [13,14]. To date, reports on the biological activities, phytochemistry and safety of many medicinal plants used in South African folk medicine against skin conditions have not been adequately explored [13]. Recently, Asong et al. [15] documented a rich pool of botanicals used by Batswana traditional practitioners for treating skin-related diseases in the Ngaka Modiri Molema District Municipality, North West Province, South Africa. Among the 80 plants identified in the study area, *Drimia sanguinea* (Schinz) Jessop (Synonym: *Urginea sanguinea* Schinz), *Elephantorrhiza elephantina* (Burch.) Skeels (Synonym: *Elephantorrhiza burchellii* Benth.), *Helichrysum paronychioides* DC (Synonym: *Gnaphalium paronychioides* Sch.Bip.) and *Senecio longiflorus* DC Sch.Bip (Synonym: *Kleinia longiflora* DC) were well-cited as the popular remedies for skin diseases. These four aforementioned plants are known for their diverse biological activities, including antimicrobial, antioxidant, toxicity and anti-inflammatory activities [16,17,18], as well as their rich phytochemical pools [16,19,20,21]. Despite the diverse biological activities of these medicinal plants, their effects against microbes implicated in common skin diseases are not well-documented. Thus, the current study investigated the pharmacological (antimicrobial and antioxidant) activities and phytochemical composition as well as the safety of four plants used in the treatment of skin diseases by Batswana traditional health practitioners in the Ngaka Modiri Molema District Municipality, South Africa.

## 2. Results and Discussion

### 2.1. Antibacterial Activity 

In the current study, the extracts obtained from the four plant species displayed varying antimicrobial activity (measured based on their minimum inhibitory concentrations, MICs) as shown in Table 1. This activity was classified as noteworthy (MIC ≤ 1 mg/mL), moderate (MIC from 1–8 mg/mL) or weak (MIC from 8–12.5 mg/mL) [22,23]. On this basis, 50% MeOH extracts were generally more potent than the PE extracts (Table 1). *Helichrysum paronychioides* (PE) and *Senecio longiflorus* (50% MeOH) were the most active extracts against *Shigella flexneri* (MIC = 0.10 mg/mL). Furthermore, 50% MeOH extracts of *Elephantorrhiza elephantina* (MIC = 0.20 mg/mL) and *Helichrysum paronychioides* (MIC = 0.39 mg/mL) had noteworthy activity against both bacterial strains used in this study. However, both bacteria had the same degree of susceptibility (MIC ≥ 6.25 mg/mL) to PE and 50% MeOH extracts of *Drimia sanguinea*. In this study, *Shigella flexneri*, a Gram-negative, was more susceptible to the extracts than the Gram-positive *Bacillus cereus*. This finding is in contrast with the common observations that Gram-negative bacteria are more resistant (relative to Gram-positive) to medicinal plant extracts due to the presence of lipopolysaccharides on their outermost membrane [24,25,26]. Three out of the four tested plants exhibited noteworthy activity against the Gram-negative *Shigella flexneri*. This is significant given the paucity of plant extracts with noteworthy activity against Gram-negative bacteria [27,28].

The plants investigated in this study can serve as potential lead candidates to explore for the control and management of skin diseases caused by Gram-negative bacteria and other related pathogens. This finding also strengthens the position of indigenous knowledge and medicinal plants in particular, in the management of skin infections caused by both Gram-positive and Gram-negative bacteria, if fully explored [29,30,31,32,33,34,35]. The moderate antibacterial activity suggests that the plants could serve as a source of natural products and antibiotics with mild to resilient antiseptic potential. The antibacterial activity of *Elephantorrhiza elephantina* has been extensively documented [36]. The authors observed noteworthy (MIC ranging from 0.08 to 0.16 mg/mL) activity of four solvent extracts against different dermatologically relevant pathogenic strains. This aforementioned finding is similar to the result in this study. *Drimea sanguinea* extracts showed moderate to weak activity with the best activity recorded at a MIC value ≥ 6.5 mg/mL, resonating with previous studies [16,37,38]. To the best of our knowledge, the antibacterial activity of *Helichrysum paronychioides* whole plants and *Senecio longiflorus* stem and leaves is being reported here for the first time.

### 2.2. Antifungal Activity

In this study, with the exception of *Helichrysum paronychioides*, the polar (50% MeOH) extracts had higher antifungal activity than the non-polar (PE) ones (Table 1). Among the tested plants, *Elephantorrhiza elephantina* (50% MeOH) extract had the most potent (MIC = 0.10 mg/mL) antifungal activity, which was exhibited against *Candida glabrata*, *Trichophyton rubrum* and *Trichophyton tonsurans*. In addition, polar and non-polar extracts of *Helichrysum paronychioides* had an MIC value of 0.39 mg/mL against *Trichophyton tonsurans*, while the non-polar ones inhibited *Candida krusei* at 0.39 mg/mL. In this study, *Trichophyton tonsurans* was the most susceptible fungal strain followed by *Candida glabrata*, while *Candida krusei* and *Trychophyyton rubrum* were the most resistant. The antifungal activity demonstrated by *Elephantorrhiza elephantina* and *Helichrysum paronychioides* is suggestive of the presence of compounds and/or possible synergistic interaction of compounds that can disrupt fungal membranes. The current finding is an indication of the therapeutic relevance of the tested plants in managing dermatological conditions of fungal origin. 

Pathogenic strains, such as *Candida albicans* and lately, *Candida krusei* and *Candida glabrata,* have been isolated from patients suffering from oral candidiasis [39]. In addition, increasing resistance to antifungal drugs especially the ‘azoles’ by pathogenic fungal strains such as resistant *Candida albicans* strains, *Candida krusei* and *Candida glabrata* has been noted [40,41]. The antifungal activity of plants from the current study, suggests that the plants can serve as a potential source of novel antifungal agents which could be promising in controlling candidiasis linked to diseases including HIV/AIDS [39]. Superficial mycosis and dermatophytosis (*Tinea corporis*, *Pityriasis versicolor*) are common infections, especially in rural areas caused by members of the *Trichophyton* genus for example, *Trichophyton rubrum* [42]. Thus, the observed noteworthy antifungal activity exhibited by *Elephantorrhiza elephantina* and *Helichrysum paronychioides* extracts warrants further investigation. *Drimia sanguinea* and *Senecio longiflorus* exhibited moderate to weak (MIC of 1.56–12.5 mg/mL) fungistatic properties. The antifungal activity of other members of *Senecio*, especially *Senecio vulgaris* against pathogenic *Candida albicans*, *Mycosporum gypseum* and *Trichophyton tonsurans* has been reported [43]. The authors observed the resistance of *Candida albicans* to extracts from *Senecio fluviatilis*, *Senecio nemorensis, Senecio pseudo-orientalis* and *Senecio racemosus*. However, in this study, *Senecio longiflorus* exhibited some degree of antifungal potency against the two *Candida* species evaluated. Similarly, the antifungal potency of members of the genus *Drimia,* notably *Drimia indica* and *Drimia elata* against *Candida albicans* and *Candida krusei* has been documented [44,45,46]. This is consistent with the result of this study, although the concentrations were at different levels. This points to the medicinal value of the plant species and members of the genera at large. Nevertheless, further investigations are needed to fully understand their mechanism(s) of activity and establish their medicinal potential. 

### 2.3. Antioxidant Activity

#### 2.3.1. 2,2-Diphenyl-1-picryl Hydrazyl (DPPH) Radical Scavenging Activity

Based on the EC_50_ values (Table 2), *Elephantorrhiza elephantina* was the most potent extract, while *Drimia sanguinea* had the least DPPH scavenging activity among the tested plants. Mpofu, Tantoh, van Vuuren, Olivier and Krause [18] also reported a high (72%) DPPH scavenging potential of *Elephantorrhiza elephantina* methanolic extracts. The positive role played by free radicals in the pathogenesis of skin infections has been documented [47]. The homeostatic balance between antioxidants and reactive oxygen species is crucial in the defense mechanisms of the skin in combating skin diseases. However, a shift in the balance may result in the production of more reactive oxygen species in the skin that will catalyze the proliferation of dermatological diseases. Available evidence indicates that consumption of natural antioxidants can ward-off the analogous health effect precipitated by oxidative damage to DNA, lipids and proteins [47]. With such a scenario, shifting to natural antioxidants is inevitable. In this study, all the plants displayed noteworthy antioxidant activity, providing scientific evidence for their utilization in traditional medicine for treating skin diseases.

#### 2.3.2. β-Carotene-Linoleic Acid Assay

In the current study, the plant extracts were able to reduce the coupled oxidation of β-carotene and linoleic acid. Antioxidant activity for the investigated plants ranged from 64.8% to 84.7% compared to BHT (used as a positive control) with 89.8% based on the test model (Table 2). *Elephantorrhiza elephantina* displayed the highest activity while *Drimia sanguinea* had the lowest antioxidant potential at the tested concentration (400 µg/mL). Generally, the antioxidant response of the extracts in the β-carotene-linoleic acid model system had a similar pattern as observed in the DPPH assay (Table 2), which also corresponded directly with the flavonoid and phenolic contents in the extracts. The positive relationship between the antioxidant activity of plant extracts and associated phenolic content has been documented [48]. However, one cannot attribute the inhibition of β-carotene bleaching and linoleic acid oxidation to the major phenolic compounds only, since there may be minor compounds with significant contribution. In this study, the observed activity may be due to synergism of both minor and major compounds. These results suggest that inclusion of the tested plants in natural products and dermatological creams may help to improve deteriorating skin conditions resulting from oxidative stress.

### 2.4. Cytotoxicity Assay

All the plant extracts (PE and 50% MeOH) displayed varying degrees (50.2–552.4 and 0.015–5622 µg/mL, respectively) of toxicity against the Vero cells (Table 3). Relative to the positive control (Doxurubicin, 10.2 µg/mL), *Drimia sanguinea* 50% MeOH extract was the most (0.015 μg/mL) cytotoxic plant, followed by *Elephantorrhiza elephantina* 50% MeOH extract (9.4 µg/mL), whereas *Senecio longiflorus* 50% MeOH extract (562.2 µg/mL) was the least toxic against the Vero cells. These results indicate that although medicinal plant usage is as old as human history with perceived indigenous efficacy and safety, there still exists an element of toxicity, which necessitates cautious and non-abusive use of medicinal plants. 

The United States National Cancer Institute (NCI) criteria for cytotoxicity of crude extracts [49] following incubation for more than 48 h, establishes that a plant extract is either safe (LC_50_ ≥ 20 µg/mL) or cytotoxic if its LC_50_ value is less than 20 µg/mL. The PE extracts of the four tested plants had LC_50_ > 20 µg/mL and may be regarded as safe for use as local remedies against skin diseases. The difference in cytotoxicity between the polar and non-polar extracts in the current study indicates that most of the compounds that may be responsible for the observed toxic effect are more soluble in polar solvents. However, the possibility that synergistic interactions among different compounds in the polar extracts are possible contributors to the higher toxicity of the polar as compared to non-polar extracts cannot be neglected. Based on the brine shrimp lethality assay, the water extract of *Elephantorrhiza elephantina* rhizome was considered credible with much potential as chemotherapeutic agent with an LC_50_ value of 42.2 µg/mL, implying its safety for consumption [50]. Similarly, Maphosa, et al. [51] indicated the lack of toxicity of the water extracts of *Elephantorrhiza elephantina* based on an acute toxicity test.

In the current study, 50% MeOH extract of *Drimia sanguinea* was the most cytotoxic plant, with less than 30% viability of the Vero cells following incubation at a concentration of 0.0075 mg/mL extract. Evidence of the toxicity of *Drimia sanguinea* has been well-documented. The plant reduced cell viability of fibroblast L929 cell lines and caused a change in morphology of chick embryo neurons [16]. In addition, more than 90% inhibition of cytokine secretion by RPMI-8226 cell lines, inhibition of DNA synthesis, 50% cell viability reduction and induction of nuclear fragmentation, as well as induction of apoptosis and incidence of human poisoning caused by the consumption of *Drimia sanguinea* have been documented [16,52]. The authors attributed this toxicity to the presence of cardiac glycosides known as bufadienolides. Based on the current study, *Senecio longiflorus* is the least cytotoxic. However, toxicity from the consumption of *Senecio longiflorus* in cattle has been reported. The condition is known as “Seneciosis” and characterized by damage to organs such as liver and bile duct [53,54]. According to Pereira, et al. [55], toxicity is often mainly attributed to two reasons. Firstly, it may be due to irrational use of botanicals and associated products which may result in bioaccumulation of potential toxic elements over time. Secondly, it can be due to the interaction between the constituents of orthodox medication and herbal-based botanicals when taken together or consecutively. 

To ensure that the antimicrobial potency was not due to factors such as the toxicity of the extracts or metabolic toxins, a selectivity index (SI) was determined for each plant extract using the MIC for all the tested pathogens (Table 4). As widely accepted, SI is a measure of impending efficacy against antagonistic effects of plant extracts. Generally, a higher SI value of a crude extract often translates that the biological (for e.g., antimicrobial) activities are potentially not due to metabolic toxins [56]. When the SI value is >1, it is considered that the antimicrobial compounds are likely different from the toxic compounds or probably the extract is more toxic to the bacteria than mammalian cells [56,57]. From the results of this study, *Senecio longiflorus* had an SI value greater than 1 for most of the screened pathogens (except *Bacillus cereus* and *Candida krusei*) implying that the antimicrobial compounds are not by-products of metabolism. *Drimia sanguinea* can be considered safe when used for treating infections caused by *Candida krusei* and *Trichophyton rubrum* given that its SI values for the microbes were greater than 1. For the other plant extracts with SI value less than 1, it is possible that there may be synergistic reactions among compounds responsible for observed activity. It is also necessary to note that in vivo efficacy and toxicity of extracts upon administration does not automatically reflect in vitro properties due to other pharmacokinetic and pharmacodynamic factors [58].

### 2.5. Total Phenolic and Flavonoid Contents

The total phenolic and flavonoid contents quantified in the four plants tested ranged from 3.5 to 64 mg GAE/g DW and 1.25–27.6 mg CE/g DW, respectively (Figure 1). *Elephantorrhiza elephantina* had the highest phenolic and flavonoid contents while *Senecio longiflorus* contained the lowest concentration of both phytochemicals. Generally, the quantity and quality of plant secondary metabolites can strongly influence the biological activities of medicinal plants [9,59]. Phenolic compounds appear in the plant kingdom in notably high amounts and vary across plant parts and species [60,61]. The antioxidant potential of several medicinal plants has been attributed to the redox potential of phenolic compounds to act as single oxygen scavengers, proton donors and reducing agents [48]. In the current study, the concentration of total phenolics detected in all the plants corresponded with the antimicrobial and antioxidant activity observed. For example, *Elephantorrhiza elephantina* (with highest total phenolic and flavonoid concentrations) had noteworthy antimicrobial activity against most of the pathogens and a high antioxidant potential (Table 1 and Table 2). Conversely, *Senecio longiflorus* had the lowest phenolic and flavonoid concentrations and exhibited moderate activity against most of the tested pathogens except for *Shigella flexneri* (MIC ≤ 0.20 mg/mL).

### 2.6. Gas Chromatography—Mass Spectrometry Analysis

The GC–MS analysis revealed the presence of many bioactive compounds in all the plants. More compounds were identified in the non-polar extracts for all the plants: *Elephantorrhiza elephantina* (105:151, polar vs. non-polar), *Senecio longiflorus* (165:185, polar vs. non-polar) *Helichrysum paronychioides* (94:198, polar vs. non-polar), and *Drimia sanguina* (169, PE only), (Appendix A). In the current study, only compounds with peak areas (A%) greater than 3% were highlighted (Table 5). Eight compounds, namely dotriacontane, benzothiazole, heptacosane, bumetrizole, phthalic acid (isomers), stigmasterol, hexanoic acid (derivatives) and eicosanoic acid were detected in all the plants, while others were specific to particular plants. In this study, the highest peak area percentage (A%) for the detected compounds was 16.8% for Diisooctyl phthalate detected in *Elephantorrhiza elephantina* extracts (Table 5 and Appendix A).

Compounds such as benzothiazole, dotriaconane, lupeol, friedelan-3-one, imidazole are known to possess diverse biological activities such as antioxidant, antimicrobial, antitumor, and antiprotozoal activities as well as a chemopreventive value [62,63]. The biological activities of these compounds are an indication of the medicinal potential of these investigated plants. Although some of the compounds possess diverse pharmacological activities, the possibilities of synergism among the different phytochemicals detected in the various plants cannot be sidelined as possible contributors to the observed activities. Some of the phytochemicals, such as lupeol, have been detected in other plant families including Myrtaceae [62]. This also points to the fact that certain compounds are conserved within the plant kingdom.

## 3. Materials and Methods

### 3.1. Plant Material Collection and Extraction

*Drimia sanguinea* (bulb), *Elephantorrhiza elephantina* (rhizome), *Helichrysum paronychioides* (whole plant) and *Senecio longiflorus* (stem and leaves) were used in the current study. These plants were identified by Batswana traditional health practitioners during an ethnobotanical survey in Ngaka Modiri Molema District Municipality, South Africa. The plants were collected between the months of September 2017 and May 2018. In order to confirm the identity of the four plants, we prepared and deposited their voucher specimens at the National Herbarium located at the South African National Biodiversity Institute (SANBI), Pretoria, South Africa.

Freshly harvested plant materials were oven-dried at 40 °C to constant dryness. The dried plant material was immediately ground to powder and stored in airtight containers in the dark, at room temperature. For the pharmacological activities, extraction was done non-sequentially using two solvents namely PE and 50% MeOH.

Each of the four plants powdered material at a ratio of 1:12.5 (g/mL) in 50% MeOH and 1:6.25 (g/mL) in PE were stirred in glass conical flasks at room temperature for 1 h and later sonicated for 1 h in ice-cold water. The extracts were filtered using Whatman No. 40 filter paper (Whatman^®^ Schleicher & Schuell, London, UK) and concentrated under vacuum using a rotary evaporator at 40 °C. The concentrates were transferred into pre-weighed glass vials and dried under a stream of air at room temperature to a constant weight and kept at 10 °C in the dark until use.

### 3.2. Antimicrobial Assay

#### 3.2.1. Microorganisms and Culture Media

Six microorganisms known for their pathogenic ability to cause skin diseases in humans were selected for the study. These American Type Culture Collection (ATCC) strains were purchased from Thermofisher Scientific, Johannesburg, South Africa. Two bacterial strains: Gram-negative *Shigella flexneri* (ATCC 12022) and Gram-positive *Bacillus cereus* (ATCC 10876), as well as four fungi: *Trichophyton rubrium* (ATCC 28188), *Trichophyton tonsurans* (ATCC 28942), *Candida glabrata* (ATCC 15126) and *Candida krusei* (ATCC 14243) were used for the in vitro antibacterial and antifungal evaluation. Sabouraud dextrose (SD) agar was used to culture *Candida krusei* and *Candida glabrata* while Yeast Malt (YM) agar (Becton Dickinson, Franklin Lakes, NJ, USA) was used to culture *Trichophyton tonsurans* and *Trichophyton rubrium.* Sabouraud dextrose (SD) broth and Yeast Malt (YM) broth were used to prepare overnight cultures for the respective fungal strains. Mueller Hinton (MH) agar and broth (Merck, Modderfontein, South Africa) were used to culture the bacteria and re-suspend the bacteria cultures overnight.

#### 3.2.2. Antibacterial Assay

The MIC values of the extracts were determined using the microtitre plate dilution technique [64]. *Bacillus cereus* and *Shigella flexneri* cultures were revived by streaking on Mueller Hinton (MH) agar plates and incubating at 37 °C for 24 h. A positive colony of each bacterium was inoculated in sterile 10 mL MH broth and incubated at 37 °C in a shaker for 24 h. The overnight cultures were diluted to a ratio of 1:100 (200 µL in 20 mL). Sterile distilled water (100 µL) was added to each well of a 96-well microtitre plate from A-H. Well A was seeded with 100 µL of re-suspended plant extract (50 mg/mL) and serially diluted two-fold down to well H, mixing thoroughly and 100 µL was then discarded from well H. Neomycin (Sigma-Aldrich, Hamburg, Germany) was used as the positive control. A two-fold serial dilution of 100 µL initial aliquot of neomycin (100 µg/mL in the first well) was performed. An aliquot of 100 µL of each bacterial strain was added to each well of the microtitre plate to give approximate final inoculums of 5 × 10^5^ colony forming units/mL. Bacteria-free MH broth, bacteria with no plant extract, 50% MeOH and PE were used as negative controls. The plates were covered with parafilm and incubated at 37 °C overnight, after which we added 50 µL (0.2 mg/mL) of *ρ*-iodonitrotetrazolium chloride (INT; Sigma-Aldrich, Hamburg, Germany) to each well. Then, the microtitre plates were incubated at 37 °C for an additional 1 h to detect growth. Generally, INT is a colourless compound that is reduced to pink to red colouration by biologically active organisms [64,65]. A colour change to pink represents bacterial growth while the well with a lack of colour change indicates inhibition of the bacteria growth. The concentration of extract in the lowest well without any colour change was recorded as the MIC of the plant extract. The experiments were performed in triplicate.

#### 3.2.3. Antifungal Assay

The antifungal potency of the plant extracts was determined using the microtitre plate dilution technique with modification [66]. Overnight culture of four fungal strains was prepared by inoculating a positive colony of each *Candida glabrata* and *Candida krusei* from YM cultured plates and *Trichophyton rubrum* and *Trichophyton tonsurans* from SD cultured plates into 5 mL of YM broth and SD broth, respectively. These were incubated at 37 ºC in a water bath while shaking. In order to obtain appropriate dilution (1:1000), 400 µL of the stock (overnight) fungal culture was mixed with 4 mL of sterile 0.85% sodium chloride (NaCl) solution. The absorbance of the culture and 1 mL 0.5 McFarland solution (0.25–0.28) was measured at 530 nm using a visible spectrophotometer. The absorbance of the fungal cultures was adjusted with sterile saline to correlate with that of 1 mL 0.5 McFarland solution at a range of 0.25–0.28. Each well of the 96-well microtitre plate was seeded with 100 µL of sterile distilled water, followed by the addition of 100 µL aliquot of plant extract (50 mg/mL), into row A of the microtitre plate and diluted two-fold serially from A to H. The reference drug amphotericin B (positive control), (Sigma-Aldrich, Germany) was prepared to a concentration of 0.25 mg/mL. An aliquot of 100 µL was serially diluted two-fold. An aliquot of 100 µL of the fungal cultures was added to each well of the microtitre plate. As negative controls, the respective extract solvents were used, fungal-free YM and SD broth, as well as the fungal cultures. The plates were covered with parafilm and incubated at 37 °C overnight, after which 50 µL of 0.2 mg/mL of INT was added and incubated further for 24 h at 37 °C. The lowest concentration of the plant extracts that was capable of inhibiting fungal growth (no color change) was recorded as the MIC of the extracts. Clear wells indicated inhibition of growth while a pink to red coloration indicated growth. The assay was conducted in triplicate.

### 3.3. Antioxidant Assay

#### 3.3.1. 2,2-Diphenyl-1-picryhydrazyl (DPPH) Free Radical Scavenging (RSA) Assay

We evaluated the free radical scavenging power of the extracts using the method by Karioti, et al. [67]. Briefly, under dim light, 15 μL of each 50% MeOH extract (tested at varying concentrations) were mixed with 735 μL methanol and 750 μL DPPH at 0.1 mM. Following incubation for 30 min at 25 °C, the absorbance of the resultant mixture was measured at 517 nm using UV-spectrophotometer (SPECORD^®^ 210 Plus Analytik, Jena, Germany). Ascorbic acid and MeOH were included as the positive and negative controls, respectively. The test was done in triplicates. We calculated the % RSA for each extract/positive control as indicated below:
RSA (%) = 100 (1 − A_E_/A_D_)
where A_E_ = absorbance of the plant extract/ascorbic acid mixture while A_D_ = absorbance of the DPPH without extract/ascorbic acid.

#### 3.3.2. β-Carotene-Linoleic Acid Assay

Based on the method by Amarowicz, et al. [68], the ability of the extract to inhibit β-carotene bleaching and linoleic acid oxidation was evaluated. Briefly, 200 μL of each extract (at 6.25 mg/mL) was added to a test-tube, which was followed by 4.8 mL *β*-carotene/linoleic acid emulsion (cocktail of 10 mg β-carotene, 10 mL chloroform, 200 μL linoleic acid, 2 mL Tween 20 and 497.8 mL water). The positive and negative controls were BHT and methanol, respectively. The test was carried out in triplicates and the final concentration of the extract/positive control was 400 μg/mL in the mixture. The absorbance reading of the mixture at 470 nm was read immediately (t = 0) prior to incubation at 50 °C for 2 h. During this 2 h incubation, the absorbance was read at 30 min intervals at the same wavelength. In order to generate the rate of *β*-carotene bleaching, the following formula was applied:Rate of bleaching (R) = ln (A_t−0_/A_t−t_) × 1/t
where, A_t−0_ = absorbance of the emulsion at t = 0 time while A_t−t_ = absorbance of the subsequent 30 min intervals.

The resultant outputs (t = 30, 60 and 90 min) were used to calculate the average rate of β-carotene bleaching, which was required to calculate the % antioxidant activity (ANT) for each extract/positive control, as indicated below:% ANT = (R_control_ − R_extract_)/R_control_ × 100
where R_control_ = Rate of bleaching of the control and R_extract_ = Rate of bleaching of the extract/positive control.

### 3.4. Cytotoxicity Assay

As described by Mosmann [69], the cytotoxicity of each extract was determined using the MTT (3-(4,5-dimethylthiazol-2-yl)-2,5-diphenyltetrazolium bromide) colorimetric assay. This was achieved by determining the effect of the extracts on the viability of African green monkey kidney (Vero, ATCC^®^ CCL-81^TM^, ATCC, Manassas, VA, USA) cells. Prior to the assay, we harvested a sub-confluent of the Vero cell with trypsin-EDTA which was subsequently centrifuged for 5 min at 200 × g. The cells (5 × 10^4^ cells/mL) were cultured on media comprising of Minimal Essential Medium (MEM) supplemented with 0.1% antibiotics (gentamicin Virbac) and 5% foetal calf serum (Highveld Biological, Johannesburg, South Africa). The experiments were conducted in a microtitre plate by seeding an aliquot of 100 µL of monolayer cell suspension into each well, a set of well containing only 100 µL of MEM was included to serve as blank. In order to enhance attachment of the cells, the microtitre plates were incubated in a 5% CO_2_ chamber at 37 °C for 24 h. Thereafter, the MEM was aspirated from the wells of the microtitre plate and replaced with 200 µL of each extract in triplicates. This was serially diluted (in serum-free MEM) to obtain varying concentrations ranging from 0.0075–0.1 mg/mL and the plates were further incubated at 37 °C in a CO_2_ chamber for 48 h.

For the experiment, we included untreated cells and doxorubicin chloride (Pfizer Laboratories) as the positive control. Immediately after the first incubation, the treated (extract/positive control) media were aspirated from the well and the cells were washed using 200 µL phosphate buffered saline. Then, media (100 µL) and MTT (30 µL) were added to each well of the microtitre plate and incubated at 37 °C for additional 4 h. Finally, we gently removed the media and avoided disturbing the MTT crystals formed in the wells. Subsequently, DMSO (50 µL) was added and carefully shaken to dissolve the MTT crystals and the absorbance of the solution was measured at 570 nm and a reference wavelength of 630 nm. The first columns of the microtitre plate with MTT and media (lacking cells) were used as a blank when measuring the absorbance of the MTT solution.

The measured absorbance values were directly proportional to the surviving cells and viable cell growth and the results were expressed as a % of the cells used in the blank. We generated the dose-response curves by plotting the % growth of the Vero cells against the log of the concentration of plant extracts. In addition, we calculated the lethal concentration (LC_50_) for each plant extract.

Based on the same units (mg/mL), the MIC values of the extracts against the six tested pathogens and the LC_50_ values of the extracts against the Vero cells were used to determine the selectivity index (SI) for each extract using the following formula:$$\mathrm{SI}\;=\;\frac{\mathrm{LC}50}{\mathrm{MIC}}$$

### 3.5. Phytochemical Analysis

Ground dried plant material (0.2 g) were dissolved in 10 mL 50% MeOH and vortexed at 2000 rpm for 2 min (VELP Scientifica Vortex, Usmate Velate MB, Italy). This was filtered at room temperature prior to use.

#### 3.5.1. Determination of Total Phenolic Content

The total phenolic content was quantified using the Folin-Ciocalteu (Folin-C) method with a modification as described by Makkar [70]. In triplicates, a reaction mixture consisting a total volume of 2 mL was made up of water (450 µL), 1 N Folin-C reagent (250 µL), 2% NaCO_3_ (1250 µL) and plant extract (50 µL). Gallic acid was used for the calibration curve.

The mixture was briefly vortexed for 6 s and incubated at room temperature for 40 min, after which the absorbance at 725 nm was recorded using a UV-spectrophotometer (SPECORD^®^ 210 plus, Analytik, Jena, Germany). The blank reaction was prepared in similar manner with 50% MeOH instead of sample. The phenolic content was reported as mg gallic acid equivalents (GAE)/g DW.

#### 3.5.2. Determination of Flavonoid Content

The flavonoid content was determined using the method described by Zhishen, et al. [71], with a modification. In triplicates, the reaction mixture of 2.5 mL was prepared by adding an aliquot of 250 µL of each plant sample diluted in 1000 µL of sterile distilled water followed by 75 µL of 5% NaNO_2_ and after 5 min, 75 µL of 10% AlCl_3_ was added. After, we added 500 µL of 1 M NaOH and 600 µL of water to obtain a mixture that was carefully vortexed and incubated at room temperature. The absorbance of the resultant mixture was recorded at 510 nm with a UV-spectrophotometer. The blank reaction mixture containing 50% MeOH instead of plant sample extract was prepared similarly. Catechin was used for plotting the calibration curve. The flavonoid concentration in the tested samples was expressed as mg catechin equivalents (CE)/g DW.

#### 3.5.3. Gas Chromatography–Mass Spectroscopy (GC–MS) Analysis

In order to determine the phytochemical profiles, PE and 50% MeOH of the four plants were subjected to GC-MS analysis, as detailed recently by Mwinga et al. [72]. In term of the capillary column, the internal diameter and film thickness were 30 × 25 mm and 0.2 µm, respectively. The carrier gas used for the analysis was ultra-high purity helium flowing at 1 mL and a speed of 37 cm/s.

For the analysis, we injected the diluted sample (1 µL) into the column with an inlet temperature of 250 °C and a splitless mode of 30 s. The initial temperate of the oven was 40 °C with a steady (at the rate of 10 °C per min with a holding time of 3 min at each increment) increment up till 300 °C. For the current experiment, we used electron ionization mode and electron multiplier/detector voltage of 70 eV (EI+) and 1750 V, respectively. The temperature of the ion source was 230 °C and 280 °C for MS transfer line. Further details were as outlined by Mwinga et al. [72].

In order to determine the phytochemicals in the different plant extracts, reference was made to the library in the National Institute of Standards and Technology (NIST). For any particular retention time, we compared the mass spectrum for each analyte with the standards at the reference library. Subsequently, we calculated the area percentage of each component by comparing individual average peak area with that of total areas.

### 3.6. Data Analysis

A non-linear regression model (curve of best fit) of a sigmoidal dose-response curve was used to calculate the EC_50_ and LC_50_ values using GraphPad Prism 5.04 software (Graphpad, San Diego, CA, USA). The data were subjected to one-way analysis of variance (ANOVA) and means were separated using Tukey’s Multiple Comparison Test.

## 4. Conclusions

In this study, we provided baseline pharmacological and phytochemical data of the four tested plants which are prescribed by Batswana traditional health practitioners for skin-related diseases. The observed antimicrobial and antioxidant activities provided the scientific rationale for the traditional uses of the investigated plants for treatment of skin diseases. In the biological assays, polar solvent (50% MeOH) extracts were generally more potent than the polar (PE) ones. Varying concentration of total phenolic and flavonoid in the plant extracts partly accounted for the observed biological activities. In term of the profiled compounds with the use of GC-MS, the presence of benzothiazole, dotriaconane, lupeol, friedelan-3-one and imidazole are noteworthy given that these compounds are well-known to possess diverse biological activities especially antioxidant and antimicrobial. Apart from *Drimia sanguinea*, which demonstrated a high degree of cytotoxicity, all the other three plants appear to be safe for consumption. However, the observed low SI values for most of the plant extracts is also of great concern and an indication of the existence of toxic compounds in varying concentrations. In addition, the issue of dosage, which can be related to the concentrations of toxic principle in the plant, must also be considered for all the tested plants. In order to fully explore the potential of these studied plants, in vivo biological assessments of the extracts and associated bioactive compounds are required to establish their pharmaceutical and pharmacognostic relevance.

## Figures and Tables

**Figure 1 plants-08-00350-f001:**
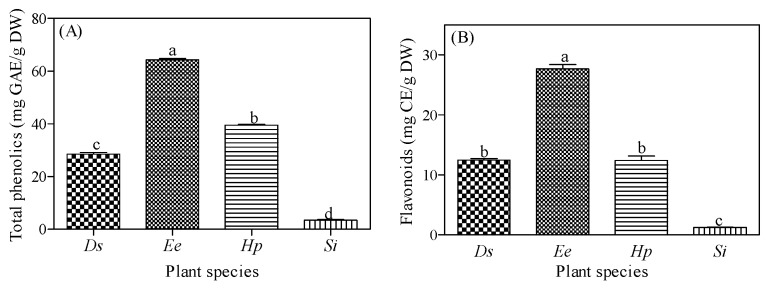
Total phenolic and flavonoid contents of 50% methanol extracts of four plants used by Batswana traditional healers to treat skin diseases. *Ds* = *Drimia sanguinea*, *Ee = Elephantorrhiza elephantina*, *Hp = Helichrysum paronychioides*, and *Si = Senecio longiflorus*. In each graph, the bars with different letter(s) are significantly (*p* ≤ 0.05) different based on Tukey’s Multiple Comparison Test. Each bar represents mean ± standard error, and *n* = 3.

**Table 1 plants-08-00350-t001:** Antimicrobial activity (minimum inhibitory concentration, MIC) of four plants used by Batswana traditional health practitioners to treat skin diseases. For the plant extracts, the values in bold are considered as noteworthy (MIC ≤ 1 mg/mL) antimicrobial activity. MeOH = methanol, PE = petroleum ether, * positive control (µg/mL) and na = not applicable.

Plant, Voucher No., Family	Plant Part	Extract Yield (% *w*/*w*)	Solvent Extract	Antibacterial Activity MIC (mg/mL)		Antifungal Activity MIC (mg/mL)			
				*Bacillus cereus*	*Shigella flexneri*	*Candida glabrata*	*Candida krusei*	*Trichophyton rubrum*	*Trichophyton tonsurans*
*Drimia sanguinea* (Schinz) Jessop **Ja004**, Asparagaceae	Bulb	7	50% MeOH	6.25	6.25	3.125	1.56	1.56	6.25
		6	PE	6.25	6.25	6.25	3.125	3.125	6.25
*Elephantorrhiza elephantina* Benth (Burch) Ja015, Fabaceae	Rhizome	22	50% MeOH	**0.20**	**0.20**	**0.10**	3.125	**0.10**	**0.10**
		6	PE	3.125	3.125	3.125	6.25	3.125	6.25
*Helichrysum paronychioides* DC. Humbert Ja037, Asteraceae	Whole plant	9	50% MeOH	**0.39**	**0.39**	6.25	6.25	6.25	**0.39**
		3	PE	1.56	**0.10**	1.56	**0.39**	3.125	**0.39**
*Senecio longiflorus* (DC). Sch.Bip Ja071, Asteraceae	Stem & leaves	10	50% MeOH	6.25	**0.10**	3.125	6.25	3.125	3.125
		4	PE	6.25	6.25	6.25	12.5	3.125	6.25
* Neomycin (μg/mL)				1.56	0.78	na	na	na	na
* Amphotericin B (μg/mL)				na	na	0.78	1.56	0.78	0.78

**Table 2 plants-08-00350-t002:** Antioxidant activity [2,2-diphenyl-1-picryhydrazyl (DPPH) free radical scavenging and β-carotene-linoleic acid models] of 50% methanol extracts of four plants used by Batswana traditional health practitioners to treat skin diseases.

Plant	Plant Part Used	DPPH (EC_50_ µg/mL)	^#^ Antioxidant (%)
*Drimia sanguinea*	Bulb	92.6 ± 4.34 ^d^	64.8 ± 1.05 ^c^
*Elephantorrhiza elephantina*	Rhizome	5.8 ± 0.46 ^a^	84.7 ± 0.59 ^a^
*Helichrysum paronychioides*	Whole plant	20.1 ± 0.42 ^b,c^	84.4 ± 0.69 ^a^
*Senecio longiflorus*	Stem and leaves	12.1 ± 0.35 ^a,b^	75.3 ± 0.25 ^b^

Positive controls, Ascorbic acid = 3.9 ± 0.18 µg/mL (for DPPH assay); BHT = 89.8 ± 0.29% (for β-carotene-linoleic acid assay). ^#^ Extracts and BHT were tested at 400 µg/mL in the β-carotene-linoleic acid assay. In each column, values (mean ± standard error, *n* = 3) with different letter(s) are significantly (*p* ≤ 0.05) different based on Tukey’s Multiple Comparison Test.

**Table 3 plants-08-00350-t003:** Cytotoxicity (LC_50_) of extracts from four plants used by Batswana traditional health practitioners to treat skin diseases. The LC_50_ (µg/mL) values represent the mean ± standard error (*n* = 3). The values in bold represent extracts considered toxic; LC_50_ < 20 µg/mL (the established cytotoxicity safety standard) and that of the positive control (Doxorubicin: 10.2 µg/mL). MeOH = methanol, PE = petroleum ether.

Plant	Part Used	Extract Type	LC_50_ (µg/mL)
*Drimia sanguinea*	Bulb	50% MeOH	**0.015 ± 0.01**
		PE	552.4 ± 48.00
*Elephantorrhiza elephantina*	Rhizome	50% MeOH	**9.4 ± 3.90**
		PE	173.5 ± 13.00
*Helichrysum paronychioides*	Whole plant	50% MeOH	24.6 ± 0.40
		PE	50.2 ± 1.80
*Senecio longiflorus*	Stem and leaves	50% MeOH	5622.0 ± 44.00
		PE	105.2 ± 79.00

**Table 4 plants-08-00350-t004:** Selectivity index of extracts from four plants used by Batswana traditional health practitioners to treat skin diseases. The LC_50_ (µg/mL) was converted to same unit (mg/mL) as the minimum inhibitory concentration (MIC). MeOH = methanol, PE = petroleum ether.

Plant Species (Plant Part)	Extract Type	Selectivity Index (LC_50_/MIC)					
		Bacterial Species		Fungal Species			
		*Bacillus cereus*	*Shigella flexneri*	*Candida glabrata*	*Candida krusei*	*Trichophyton rubrum*	*Trichophyton tonsurans*
*Drimia sanguinea* (Bulb)	50% MeOH	0.0002	0.0002	0.0005	**9.6**	**9.6**	0.0002
	PE	0.09	0.09	0.09	0.18	0.18	0.09
*Elephantorrhiza elephantina* (Rhizome)	50% MeOH	0.047	0.047	0.094	0.003	0.047	0.047
	PE	0.05	0.05	0.05	0.3	0.05	0.3
*Helichrysum paronychioides* (Whole plant)	50% MeOH	0.06	0.06	0.004	0.004	0.004	0.06
	PE	0.03	0.5	0.03	0.1	0.01	0.1
*Senecio longiflorus* (Stem and leaves)	50% MeOH	0.9	**56.2**	**1.8**	0.9	**1.8**	**1.8**
	PE	0.4	0.4	0.4	0.008	0.03	0.4

A selectivity index value greater than 1 (written boldly) means that the extract is more toxic to the microbes than to mammalian cells. The greater the selectivity index value, the safer the plant extract.

**Table 5 plants-08-00350-t005:** Compounds detected by Gas chromatography–mass spectrometry (GC-MS) analysis with peak area greater than 3% from polar and non-polar extracts of the four plants used by the Batswana traditional health practitioners to treat skin diseases. A% = Area percentage, MF = Molecular formula, SI = Similarity index and DT = Detection time.

Name of Compound	A% ≥ 3	MF	SI (%)	DT(s)
A. *Drimia sanguinea* (PE)				
(1) Pentanoic acid	3.8	C_5_H_10_O_2_	91.2	171.5
(2) n-Hexadecanoic acid	5.9	C_16_H_32_O_2_	92.4	972.0
(3) 1-Nonadecene	5.1	C_19_H_38_	76.1	977.1
(4) Hexadecanoic acid, ethyl ester	5.0	C_18_H_36_O_2_	89.5	977.5
(5) Diisooctyl phthalate	5.2	C_24_H_38_O_4_	92.1	1302.7
(6) α-Sitosterol	3.5	C_29_H_50_O	89.8	1657.9
B1. *Elephantorrhiza elephantina* (PE)				
(7) Diisooctyl phthalate	16.8	C_24_H_38_O_4_	92.6	1304.5
(8) Pregnenolone	7.9	C_21_H_32_O_2_	54.5	1657.3
(9) α-Sitosterol	7.8	C_29_H_50_O	77.1	1659.3
(10) Lupeol	7.9	C_30_H_50_O	90.7	1686.6
(11) Cycloeucalenol acetate	8.2	C_32_H_52_O_2_	77.3	1687.7
(12) Unknown 2	3.6	C_24_H_28_O_3_S	46.8	1684.6
B2. *Elephantorrhiza elephantina* (50% MeOH)				
(13) Pentanoic acid, 2-methyl-, anhydride	7.7	C_12_H_22_O_3_	80.3	233.4
(14) Pentanoic acid, 4-oxo-	3.5	C_5_H_8_O_3_	92.6	235.9
(15) 1H-Imidazole-4-ethanamine, α-methyl-	3.1	C_6_H_11_N_3_	60.5	237.2
(16) Benzothiazole	3.0	C_7_H_5_NS	96.2	327.0
(17) Carbonic acid, but-3-yn-1-yl heptadecyl ester	4.3	C_22_H_40_O_3_	64.3	613.6
C1. *Helichrysum paronychioides* (PE)				
(18) Methyl 2,4,6-trihydroxybenzoate	3.1	C_8_H_8_O_5_	72.5	899.4
(19) 2-Chloroethanol, triisobutylsilyl ether	6.7	C_14_H_31_CIC	62.6	1152.7
(20) (1-Cyclohexylmethyl-3-methylbut-2-enylthio)benzene	6.7	C_18_H_26_S	56.1	1153.3
(21) Benzenamine, 2-iodo-	5.2	C_6_H_6_IN	65.2	1172.8
(22) 2(3H)-Benzofuranone, 3α,4,5,6-tetrahydro-3α,6,6-trimethyl-	9.0	C_11_H_16_O_2_	64.7	1199.3
(23) 4,5,6,7-Tetrahydro-benzo[c]thiophene-1-carboxylic acid allylamide	8.6	C_12_H_15_NOS	60.8	1200.0
(24) 3-Buten-2-one,4-(3-hydroxy-6,6-dimethyl-2-methylenecyclohexyl)-	3.9	C_13_H_20_O_2_	60.6	1211.2
(25) Unknown 6	3.7	C_17_H_17_N_3_O	29.5	1211.5
C2. *Helychrysum paronychioides* (50% MeOH)				
(26) Methyl 2,4,6-trihydroxybenzoate	4.5	C_8_H_8_O_5_	72.5	899.4
(27) Falcarinol, trimethylsaline	5.8	C_20_H_32_OSi	50.1	1150.4
(28) 1-(5-Hexyl-2,4-dihydroxyphenyl) ethanone	5.8	C_14_H_20_O_3_	62.8	1151.0
(29) Unknown 4	5.4	C_12_H_14_N_4_O_4_	44.4	1152.3
(30) 2,5-Cyclohexadiene-1,4-dione, 2,5-bis(1,1-dimethylpropyl)-	3.3	C_16_H_24_O_2_	64.5	1192.3
(31) Pyrimidine-5-carboxylicacid,1,2,3,4-tetrahydro-6-methyl-2-oxo-4-(2-thienyl)-, isopropyl ester	5.3	C_13_H_16_N_2_O_3_S	62.4	1197.6
(32) 4,5,6,7-Tetrahydro-benzo[c]thiophene-1-carboxylic acid allylamide	5.4	C_12_H_15_NO	63.7	1197.8
(33) 5H-Benzo[b]pyran-8-ol,2,3,5,5,8α-pentamethyl-6,7,8,8α-tetrahydro-	3.2	C_14_H_22_O_2_	60.4	1209.5
(34) Olivetol, [Tert-butyl (dimethyl)silyl]	5.9	C_17_H_30_O_2_Si	56.0	1240.2
(35) 4,5,6,7-Tetrahydro-benzo[c]thiophene-1-carboxylic acid allylamide	5.9	C_12_H_15_NO	62.0	1241.5
(36) 4′-hydroxy-2′,6′-dimethoxy-3′-(3-methyl-2-butenyl)-acetophenone	4.3	C_15_H_20_O_4_	55.9	1282.2
(37) 4,5,6,7-Tetrahydro-benzo[c]thiophene-1-carboxylic acid allylamide	4.3	C_12_H_15_NO	55.7	1284.3
(38) 4,5,6,7-Tetrahydro-benzo[c]thiophene-1-carboxylic acid allylamide	4.2	C_12_H_15_NO	62.0	1285.0
(39) 2,5-cyclohexadien-1-one,2,6-bis(1,1-dimethylethyl)-4-hydroxy-4methyl-	3.5	C_15_H_24_O_2_	60.1	1296.4
D1. *Senecio longiflorus* (PE)				
(40) Dotriacontane	13.6	C_32_H_66_	94.7	1604.0
(41) 6 βBicyclo [4.3.0] nonane, 5β-iodomethyl-1β-41) isopropenyl-4α,5α-dimethyl-,	3.0	C_15_H_25_I	68.3	1676.0
(42) Lupeol	7.4	C_30_H_50_O	88.7	1680.2
(43) 1,3,6,10-Cyclotetradecatetraene, 14-isopropyl-3,7,11-trimethyl-	7.8	C_20_H_32_	79.5	1680.6
(44) 9,19-Cyclolanostan-3-ol, acetate, (3β)-	4.2	C_32_H_54_O_2_	69.7	1688.9
(45) Heptacosane	3.6	C_27_H_56_	90.1	1689.8
(46) Lup-20(29)-en-3-one	4.0	C_30_H_48_O	82.1	1692.1
(47) Lupeol	6.8	C_30_H_50_O	86.8	1697.5
(48) 1,3,6,10-Cyclotetradecatetraene, 14-isopropyl-3,7,11-trimethyl-	8.5	C_20_H_32_	82.8	1698.4
D2. *Senecio longiflorus* (50% MeOH)				
(49) 1-Hydroxymethyl-2-methyl-1-cyclohexene	11.1	C_8_H_14_O	70.1	272.3
(50) 4H-Pyran-4-one, 2,3-dihydro-3,5-dihydroxy-6-methyl-	11.1	C_6_H_8_O_4_	87.9	272.6
(51) 4-Hydroxy-4-methylhex-5-enoic acid, tert.-butyl ester	10.0	C_11_H_20_O_3_	68.7	274.0

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
