# Peer review of "Antimicrobial Activity, Antioxidant Potential, Cytotoxicity and Phytochemical Profiling of Four Plants Locally Used against Skin Diseases"

_plants, 2019, doi:10.3390/plants8090350_

Round 1

Reviewer 1 Report

Dear Authors,

The article is original and very important in the field of studies. Nowadays is actively developing searching of specific biological active compounds with antifungal and antimicrobial effects. 

After careful read of Introduction part and whole body of article I would like to admit that would be better to say about studies which are searching of specific biological active compounds with antifungal and antimicrobial effects. I would like to admit that its not just flavonoids and phenolics. Its better to say more common specific secondary metabolites. For example, specific naphtodynathrones also shown antifungal effects against Trichophyton rubrum and Candida albicans. Please see references

https://doi.org/10.1080/13880209.2016.1211716 doi: 10.3389/fphar.2018.00382

I would check is possible to make conclusion on the presented data how Antimicrobial Activity, Antioxidant Potential and Cytotoxicity are connected with presence phenolics or some specific phenolic compounds (from GC-MS analysis) in the studied plants.

Abstract must be described more detailed and shorten to show essence of study. 

Author Response

The article is original and very important in the field of studies. Nowadays is actively developing searching of specific biological active compounds with antifungal and antimicrobial effects. 

After careful read of Introduction part and whole body of article I would like to admit that would be better to say about studies which are searching of specific biological active compounds with antifungal and antimicrobial effects. I would like to admit that its not just flavonoids and phenolics. Its better to say more common specific secondary metabolites. For example, specific naphtodynathrones also shown antifungal effects against Trichophyton rubrum and Candida albicans. Please see references

https://doi.org/10.1080/13880209.2016.1211716 doi: 10.3389/fphar.2018.00382

RESPONSE

We have read the suggested literature (Sytar O, Švedienė J, Ložienė K, Paškevičius A, Kosyan A, Taran N (2016) Antifungal properties of hypericin, hypericin tetrasulphonic acid and fagopyrin on pathogenic fungi and spoilage yeasts. Pharmaceutical Biology 54 (12):3121-3125. doi:10.1080/13880209.2016.1211716)

Thank you very much for your comments. We do accept that there are diverse groups of plant metabolites that have antifungal and antibacterial activity. However, we have made specific reference to phenolic and flavonoid in order to align with the focus of the study that evaluated the concentrations of these aforementioned metabolites in the plant extracts. Hence, we tried to establish a correlation between these specific metabolites with references existing literature during the conceptualisation of the study.

I would check is possible to make conclusion on the presented data how Antimicrobial Activity, Antioxidant Potential and Cytotoxicity are connected with presence phenolics or some specific phenolic compounds (from GC-MS analysis) in the studied plants.

RESPONSE

In the conclusion, we have revised the content to provide argument on the potential link among the studied biological activities and cytotoxicity as well as the phenolics and specific chemicals quantified/identified in the plants. We have revised the content from 102 words to 200 words.

Abstract must be described more detailed and shorten to show essence of study. 

RESPONSE: We have revised the abstract and shorten the number of words from ‘305’ to ‘260’.

RESPONSE: As suggested above, we have improved the following sections: Introduction and conclusion. In term of the research design, no addition changes were made as we strongly believe the current content is well-designed and nothing has been excluded. These aforementioned changes are highlighted in red.

Reviewer 2 Report

The manuscript consists of the investigation into biological activity of four plant species which are used for medicinal purposes by South African natives. Certainly there is a niche for studies of biologically-active plants which are successfully used by the local folk. The report is original and highly interesting. The research is well prepared, and clearly presented. The scientific results presented by authors are properly referenced and discussed. Only a minor spelling check is required. The manuscript is worth-publishing in Plants Journal and I gladly recommend the acceptance.

Reviewer 3 Report

The paper, entitled " Antimicrobial activity, antioxydant potential, cytotoxicity and phytochemical profiling of four plants locally-used against skin diseases " is in the topic of the journal. It provides interesting information on the pharmacological activities of a polar and a non polar extracts from four different medicinal plants used by indigenous people for treating skin diseases.

The part materials and Methods is well-described and detailed.

The results are properly written and well-discussed. The conclusion, admittedly short, envisages perspectives on in vivo activities.

Just one remark, the latin names of the different plants must be written in italics (lines 115,116, 118 120, 121).

For these reasons I accept the paper for publication.
